# Water Conserving Message Influences Purchasing Decision of Consumers

**Melinda J. Knuth** [1], **Bridget K. Behe** [2,*], **Patricia T. Huddleston** [3], **Charles R. Hall** [4], **R. Thomas Fernandez** [2] **and Hayk Khachatryan** [1]

[1] Food and Resource Economics Department, University of Florida, Apopka, FL 32703, USA; melindaknuth@ufl.edu (M.J.K.); hayk@ufl.edu (H.K.)

[2] Department of Horticulture, Michigan State University, East Lansing, MI 48824, USA; fernan15@msu.edu

[3] Department of Advertising and Public Relations, Michigan State University, East Lansing, MI 48824, USA; huddles2@msu.edu

[4] Department of Horticultural Sciences, Texas A&M University, College Station, TX 77845, USA; c-hall@tamu.edu

\* Correspondence: behe@msu.edu; Tel.: +1-517-353-0346

**Abstract:** As more of the U.S. population urbanizes, freshwater resources will become more partitioned and scarcer. Live plants need water to become established and survive, but water demands vary by taxa. Additionally, outdoor household water use is becoming a greater target for watering restrictions, particularly landscape irrigation. Yet, how important is water conservation messaging in the context of a plant purchasing decision for outdoor plants? A ratings-based conjoint analysis of the water message, plant type, price, and plant guarantee was conducted using data from 288 subjects from three locales who rated their purchase intention to buy a plant from a retail merchandising display while using an eye-tracking device. Subjects were clustered by state of residency and, separately, their conjoint utility scores. Results indicate that water-related messaging does play a positive role in purchase intention. Residents of states who frequently experienced drought rated a water-saving message higher compared to residents of states who experienced relatively little drought. For some consumer groups, such as plant buyers, water savings are important and sought after. Green industry stakeholders should be aware of their region's drought history and help develop point of purchase information to include water conservation messaging in retail displays.

**Keywords:** purchase decision; landscape plants; signage; drought; eye-tracking

## 1. Introduction

The scarcity of water due to the increase in urbanization has led to the separation of water needs into categories such as outdoor or indoor, business or household, etc. [1]. While indoor water usage appears stable from season to season, outdoor water usage is more varied, considered a discretionary use, and is subject to regulation surrounding its usage [2,3]. Outdoor water application constitutes approximately 30% of water used in and around residential homes [4]. Outdoor water use is usually determined by seasonal need, garden type and importance, social norms, and landscape area size. Therefore, highly visible landscape, lawn, and garden irrigation are frequently targeted for mandatory restrictions and regulation. Increased regulations generally result in an increase in direct and indirect costs, and an indirect cost of watering restrictions is that they often result in landscapes being altered to require less water. Fortunately, water costs must be weighed against the benefits of water conservation. Convincing consumers that the costs of water conservation yields environmental benefits is of utmost importance. Yet, water messaging has not been thoroughly investigated to understand its influence on consumer water conservation behavior and its subsequent impact on purchase intentions.

Thus, our research question is: how does a water conservation message in the retail environment affect consumers' purchase intention for outdoor plants? To investigate this question, we employed two common consumer research methods: a conjoint design supplemented with a survey questionnaire. During the conjoint portion of the study, we utilized an eye-tracking device to measure visual attention to the water message as well as other areas of interest (e.g., price, plant material, and product guarantee length). The survey included questions regarding the subjects' landscape activities, plant expertise and knowledge, and involvement with plants. The goal was to integrate consumer responses with gaze metrics to understand how water conserving messages influence purchase intentions for live plants. Analyses were conducted both by state of residency and on consumer groups derived from an agglomerative cluster procedure using subjects' utility scores produced from the conjoint analysis.

*1.1. Homeowner Perceptions of Water Conservation in the Landscape*

In ornamental plant production, water use and recycling effectiveness begin with plant selection (e.g., lower water usage plants) and adoption of tools and techniques for water conservation [5]. For both growers and consumers alike, their decisions to conserve water may be influenced by pre-established information and attitudes, social and regulatory pressures, and environmental factors, including climate change and outdoor space availability [5].

Homeowners are more concerned with long-term drought effects than other weather patterns, such as heavy rains [6]. Additionally, household water consumption decreases under differing water conditions, including drought [7]. Scant work has been done investigating drought influences on outdoor plant purchases. In one study, participants who accurately perceived themselves to be in a drought situation had a more accurate assessment of their confidence and knowledge in conserving water in their outdoor space (i.e., they were consciously competent.) [8]. Additionally, Behe et al. [9] and Knuth et al. [10] found that individuals who scored high on plant expertise and had a high level of involvement with plants also had greater involvement in water conservation, and tended to be more engaged in water conservation activities.

Demographically, low water users (conservers) tend to be older, less affluent, and more likely female [11,12]. Water consumption increases linearly with income up to $100,000, at which point the households practice greater water conservation and individuals are more likely to use drought tolerant plants in their landscape [6,13–16]. There is evidence that increased general knowledge and education are more directly linked to conservation adoption compared to other demographic influences [12,17]. Thus, water conservation messages may positively resonate with plant purchasers and demographic characteristics may also influence the effect.

*1.2. Water Messaging in Signage*

Water conservation messaging on retail signage is relatively novel. Knuth et al. [8] conducted a national survey in 2016 investigating drought influences on outdoor plant purchases. They found that consumers who were in drought, whether they knew it unconsciously or consciously, had different landscape irrigation habits based on their perception of drought status. Furthermore, subjects placed greater relative importance on the water source used during plant production (e.g., fresh, recycled, or blended water sources) over water use in the landscape (i.e., watering in the first season versus multiple seasons) for both herbaceous and woody perennial plants [8]. Additionally, consumers preferred growers who used fresh water over recycled water, and least preferred a blend of fresh with recycled in perennial and shrub production. Thus, water source appears important in the context of plant purchase decisions.

How people react to information about water reuse is thought to depend to a great extent on the information processing experience [18]. Some studies have shown that providing information, such as the risks and benefits of using recycled water or descriptions of water treatment processes, has positive impacts on water reuse [18,19]. McClaran et al. [20] found that when labeling water as "recycled" (versus "reclaimed"), "recycled" was perceived as less risky, a finding which was largely

attributed to "reclaimed" water eliciting a "yuck" response. In that same study, subjects were 40% less likely to directly use and 60% less likely to indirectly use reclaimed water compared to recycled water. Yet, using a priming message indicating that the water was recycled from a production nursery compared to a residential area reduced the perceived risk level [21].

Another study showed that plant purchasers paid more attention to water saving information on signage if they practiced water conservation in their home, and paying more attention to that message increased the participant's likelihood to buy a plant [22]. However, the messaging had no influence on consumers who practiced little to no water conservation. Additionally, participants from Florida, a state that has experienced some levels of drought, had greater sensitivity to water conserving messaging than consumers from states that experience little to no drought. In a study that compared assertive (e.g., "You must use less water") and suggestive (e.g., "It is worthwhile to save water") messaging, the use of a suggestive message had a greater effect on influencing water conserving habits than did an assertive message [23]. In an eye-tracking study that compared garden and retail signage for different environmentally friendly plant production methods, subjects preferred plants grown using environmentally friendly production methods (versus conventional). "Water conserving practices" signage held consumers' gaze for a similar length of time as "sustainability" signage, indicating subjects may be more familiar with these terms and found them easier to comprehend [24]. Water conservation messaging on retail signage may facilitate plant purchases.

### 1.3. Visual Information Processing

Attention to visual information in a purchase decision task implies an awareness of the stimuli in the conscious mind, which is driven by both top-down and bottom-up processes [25,26]. Bottom-up processes occur when important information from the external world attracts our attention, while top-down processes draw attention to information based on our (internal) knowledge, beliefs, expectations, and goals [26]. Attention drawn to a sudden loud noise, siren, or flashing lights is the result of a bottom-up process, while finding a friend's face in a crowd is the result of a top-down process. Wedel and Pieters [25] found that both top-down and bottom-up processes affected consumer behavior.

The brain processes visual stimuli two ways: by object processing (identification) and spatial processing (egocentric perspective) [27]. Visual information acquisition happens with no conscious awareness, allowing individuals to function without constantly evaluating their environment [28]. Yet, individuals can deliberately transform this spatial processing, too, like communicating in dynamic settings with other people or simulating the use of a product present in an advertisement, including pictures and text. For example, if a hose or a watering can is present in an advertisement for a plant, the consumer could cognitively transform the images to an implication that the plant needs a lot of water. Additionally, individuals perceive cues in their environment where information is cognitively gathered and used while making a decision [29]. From previous studies, there are many known influential decision cues in the retail environment, including price, brands, and packaging [27].

Visual attention influences product selection. Behe et al. [30] reported that consumers visually found their most important information cue in a purchase decision (e.g., price, production method, or plant) faster compared to the other visual cues. For example, price-oriented consumers found the price information on the sign faster than production-oriented consumers [30]. Visual attention time correlates positively with purchase intention, such that increased gaze fixation time on certain product attributes indicates an increased purchase likelihood [30–34]. Gidlof et al. [34] reported, "the very act of looking longer or repeatedly at a package, for any reason, makes it more likely the product will be bought." Furthermore, retail sign complexity influences visual attention. For example, study participants viewing garden center signs fixated more (by fixation counts) on signs with moderate visual complexity compared to low or high complexity signs, and had higher buy ratings [35]. This was perhaps explained by greater processing fluency of the moderate complexity signs (compared to highly complex signs), but containing more information compared to the low complexity signs, which contributed little to the purchase decision. Other top-down and bottom-up cues guide visual

attention. Khachatryan et al. [36] observed that subjects who scored high on the buying impulsiveness scale fixated less on point-of-purchase information and more on the product (live plants). The use of a red colored font (versus black font) led to a longer gaze length and captured attention faster [37]. Additionally, in that same study, displaying the price discount as a percentage was more preferred to "Buy-3-Get-1 Free" or as a specific dollar value, most likely due to the ease of cognitive processing. Visual attention to water conserving messages may play a role in purchase intentions.

### 1.4. Plant Guarantees

Over a century ago, L.L. Bean instituted a 100-percent product guarantee, considered by many as setting the highest standard for customer satisfaction [38]. At the time, those entrepreneurs hoped to gain the trust and eventual purchases of consumers new to the company with the offer of a money-back guarantee. To some, L.L. Bean's actions may be considered one of the first large-scale experiments to reduce consumers' perceived risk. One definition of risk [39] is, "the chance of loss . . . or the degree of probability of such loss." Derbaix [40] characterized four types of consumer risk: time, physical, psychological, and financial. A money-back guarantee directly addresses the financial risk of buying a live plant, and may alleviate some of the psychological risk (e.g., dissatisfaction or regret) [41].

Plants are live products, and their failure or death may cause the purchaser to experience negative consequences, including product dissatisfaction or regret. Live products are just as likely (or perhaps more likely because they are alive) to fail compared to non-living products. Behe and Barton [42] reported that the presence of money-back guarantees on rooted plants increased consumer satisfaction. Rihn et al. [43] showed that the presence of a guarantee reduced perceived risk and improved consumers' experience with floral products. The "Guarantee Seekers" segment in that study were more likely to choose cut flower arrangements with money-back guarantees, while the other groups were interested in both guarantees and floral longevity indicators [43]. Behe and Fry [44] showed that the presence of plant guarantees increased purchase intentions and reduced perceived product loss risk for the purchaser. Plants that require less irrigation in the landscape may contribute to the perception of risk by the purchaser, and the presence of a guarantee may mitigate some of the perceived risk.

### 1.5. Pricing and Sales

Retailers may communicate "sale" prices in order to attract visual attention and stimulate product sales because what goes unseen goes unsold. Communicating price as "regular" and "sale" has been studied through the lens of comparative pricing [45]. Some research supports the notion that consumers evaluate prices relatively and not absolutely [46,47]. For example, an item with a regular price of $99 and a sale price of $49 is likely to be perceived as a 50% discount. However, a regular price of $20 and a sale price of buy-3-get-1-free may require more cognitive effort to realize that that is a 25% discount. Grewal et al. [48] showed that for price discounts, the subject's internal reference price and their perception of brand quality greatly affected the perceived value of a product, and that perceived value positively influenced likelihood to buy. Additionally, purchase intentions increase with discounts on products with a high brand reputation, whereas purchase intentions are less predictable for products with lower brand reputation [49].

However, there is some research that shows consumers evaluate a price absolutely. How the price is "framed" influences sale perceptions [50]. In other words, the way the price is stated (as a percentage off versus a dollar amount) may affect purchases. Monetary cash discounts are preferred over freebies, and discounting preferences (e.g., freebies versus discounts) vary little by age [51], but multiple unit pricing versus percentage discounts has not been well-investigated. Coulter and Norberg [47] showed that greater perception of price discounts leads to greater value perceptions, but the physical distance between two prices influences the magnitude of the value perceptions. In other words, prices separated by great physical distance also had a greater perceived value difference.

Behe et al. [37] investigated the role of "sale" font size and color, in addition to sale price, on purchase intentions of live plants. They showed that price was the second most important product

attribute relative to plant type, followed by "sale" font color and size and sale price representation (e.g., regular price, percentage off, or multiple unit sale pricing). They also reported a synergistic effect of "sale" font size and color such that when a red and larger font was used to communicate the word "sale", it was noticed faster compared to a black font that was a similar or smaller size.

In the present study, we examine the combination of a water conservation message, plant guarantee, and price on the purchase intention of live plants in a retail display. The goal of this investigation is to evaluate the importance of a water conservation message when taken into context with other external cues, such as price, plant guarantee, and plant type.

## 2. Materials and Methods

To facilitate transparency of data collection and analyses, the survey instrument, ethics committee approval documents, data, and analytic code with results are available in the Open Science Foundation virtual Supplementary Materials (Supplementary Materials S1–S6) linked here: https://osf.io/yvd7g/ ?view_only=9ddfa79feff14fa2bae7e5dd06633589.

### 2.1. Procedures

We developed a lab experiment following widely accepted market research protocols to ensure a greater degree of accuracy and speed of data collection, while reducing human error and survey expenses [52–54]. Both the survey instrument and protocol were approved by the university committees involving research with human subjects (MI Study00000196; TX IRB2018-0108M; refer to Supplementary Materials S1 and S2 in the Open Science Foundation (OSF) repository). The survey was administered from 11 to 12 September 2018 in Florida, 2 to 4 October 2018 in Texas, and 30 October to 4 December 2018 in Michigan. For statistical power, our goal was to obtain at least 75 respondents from each state, and that at least one state would be in a drought condition and at least one not in a drought condition (an a priori power test was conducted to determine the required sample size (effect size = 0.50; alpha = 0.05; and power = 0.80)). We collected data from subjects at three geographically disparate locations in Michigan, Florida, and Texas using a largely non-student sample of individuals ≥ 18 years of age recruited from each surrounding community. Texas ($n$ = 106) was in a drought condition during data collection, but neither Michigan ($n$ = 102) nor Florida ($n$ = 80) were in a drought condition during data collection.

Upon arrival at the testing location, subjects were asked to read and sign an informed consent form and were paid a $10 incentive. Subjects then proceeded to the eye-tracking portion of the study, followed by the survey questionnaire. Subjects were seated at an eye-tracking station and had the equipment calibrated to their eye movements [55]. Subjects independently proceeded through the stimuli. The study began with self-paced, on-screen instruction and practice slides. The eye-tracking camera was mounted on the computer monitor, and the scenario images were located centrally on the screen. Each stimulus was preceded by a 2 s bull's eye to reposition the subject's gaze randomly to one of the four corners of the screen to avoid central gaze bias. Each stimulus slide consisted of an image with an 11-point likelihood to buy (LTB) rating scale at the bottom of the screen (Figure 1) [56]. The LTB ranged from 0 = No chance to 10 = Certain. When the subjects had finished rating the stimuli (stimuli discussed more in Section 2.2), they proceeded to the sociodemographic survey. Once the survey ended, the subjects were notified that the study was complete and left the laboratory.

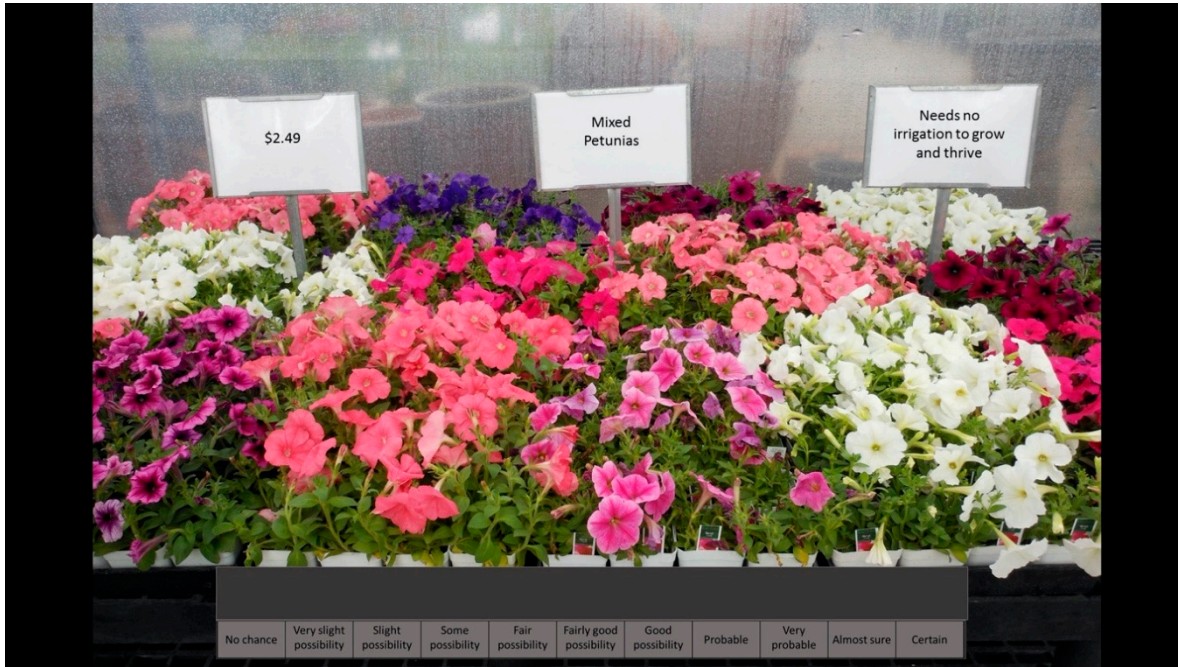

**Figure 1.** Example of the simulated retail merchandising display that was projected on a computer screen that subjects viewed with the attribute levels: annual plants, regular price, no plant guarantee, and needs no irrigation; 11-point Juster scale on the bottom.

### 2.2. Stimuli

A ratings-based conjoint study was developed to analyze how the different water messages, position in the display, price, guarantee, and type of plant affected purchase intentions. We employed a combination of product attributes and levels that represented two plant types (annuals or woody perennial shrubs), three price aspects (regular price, buy-3-get-1-free, and 25% off regular price), a plant guarantee (none, 30 days, and 90 days for annual plants and none, six months, or one year for shrubs), and water requirement messaging (needs irrigation after plant establishment or needs no irrigation after plant establishment) for a $2 \times 3 \times 5 \times 2$ design (Table 1). However, to reduce participant fatigue, we showed each subject 36 of the 60 possible combinations, yielding a fractional factorial design. The 36 images were established as the minimum required combinations for analyses, but without correlation. Price messages were established through conversations with industry professionals and were appropriate for all three states. Each digital image consisted of three blank signs, which were modified based on the conjoint design combinations and LTB rating scale at the bottom of the screen. We randomized the price message location (left, middle, or right sign), then placed the plant identification and water message systematically on the remaining two signs (moving from left to right).

The 36 stimuli images were incorporated into the Tobii X1 Light Eye Tracker (Tobii Technology, Stockholm, Sweden) software at the Michigan and Florida locations, and into the iMotions Biometric Research Platform (iMotions A/S, Copenhagen, Denmark, 2018) at the Texas location. On average, the subjects viewed each stimuli for 13.36 sec (Standard Deviation (S.D.) = 13.23 s), with a range of 3 s to 3 min and 58 s.

**Table 1.** Attributes and levels within the conjoint analysis.

| Attributes | Levels |
|---|---|
| Plant | Annuals |
| | Shrubs |
| Water Message | Needs Irrigation |
| | Needs No Irrigation |
| Price Type | Regular Price |
| | 25% Off of Regular Price |
| | Buy-3-Get-1-Free |
| Plant Guarantee | Blank |
| | 30-Day |
| | 90-Day |
| | 6 Month |
| | 1 Year |

*2.3. Analyses*

The part-worth values identified in the conjoint analysis are useful in segregating a sample into smaller clusters or market segments. Kaufman and Rousseeuw [57] stated that cluster analysis, "is the art of finding groups in data" (page 1). Using all of the part-worth utility values generated in the conjoint analysis, we conducted an agglomerative cluster analysis using SAS software (version 25, Cary, NC, USA) PROC CLUSTER, saving cluster membership for comparisons. A dendrogram and dissimilarity plot were utilized to determine how many clusters were appropriate. We conducted a hierarchical cluster analysis to identify homogeneous consumer groups.

Principal component analyses are used to describe the strength and direction of correlated variables in terms of their potential to quantify unobservable constructs [58]. The values that emerge show the interdependencies between observed variables, which can be collapsed into a smaller set of components. The key result of a principal component analysis is the independent variables' association with an indirectly measured construct or component. We used SAS to conduct two separate principal component analyses: landscape and plant importance, and plant expertise and involvement. The landscape and plant importance scales were taken from Syme et al. [53], Behe et al. [9], and Knuth et al. [10]. The plant expertise and involvement scales were taken from Behe et al. [9] and Knuth et al. [10]. In each analysis, we retained items with factor loadings ≥ 0.500 [58]. The number of factors was determined by evaluating the "elbow," or a leveling off in the eigenvalues, as well as the significant drop in variance in the initial principal component analysis [59]. "Load" or "loading" is the terminology used in principal component analyses to indicate the mean value for each item (question), being the highest among all of the mean values for that item when testing for linear combinations [59,60]. Solutions (component values) with a Cronbach's alpha level of ≥0.7 are considered to have a strong measure of internal consistency or validity [61]. Additionally, solutions with a Kaiser–Meyer–Olkin measure < 0.70 are considered acceptable (less than are considered mediocre or less quality) [62].

What goes unseen goes unsold, thus, the use of eye-tracking devices has become more commonplace in consumer research [30,35–37]. Within a scene or display, areas of interest (AOIs) are identified, and visual metrics can be extracted by AOI. The metrics were used in subsequent analyses to determine what information or AOI attracted and held the visual gaze. In this study, each sign was a separate AOI, and the plants constituted a fourth AOI. While many visual metrics can be extracted from eye-tracking devices, the three most common are the first glance or time to first fixation (TTFF), number of glances in that AOI or fixation count (FC), and gaze duration within an AOI or total fixation duration (TFD). We extracted TTFF and TFD, which were measured to one-hundredth of a second, and FC.

We compared visual metrics with purchase intentions for respondents by the state in which they lived. We tested for visual metric and LTB rating differences between the states and the clusters on demographic characteristics, conjoint utility scores, and relative importance values with regression

and generalized linear model (GLM) procedures using Stata software (version 16, College Station, TX, USA). To compare water conservation knowledge level and behavioral questions, two principal component analyses were conducted using SAS. The plant knowledge quiz included ten questions designed to measure the subject's depth of knowledge in horticulture practices and facts (refer to Supplementary Materials S7).

## 3. Results

### 3.1. Demographic Characteristics

The overall sample was 68% female (Table 2). Michigan had the greatest number of female subjects (72%), while Florida had the fewest (63%). On average, 68% of the sample identified as White. Texas had the highest ethnic diversity, with 50% identifying as non-white ethnicities: African American (6.6%), Hispanic (18.9%), Asian (15%), and the remainder identified as other ethnicities (8%). Florida had the second highest ethnic diversity, while Michigan had the least (70.6% White). Fifty-seven percent of the subjects earned a bachelor's degree or greater. Comparatively, Michigan had the lowest percentage with a bachelor's degree or greater (48%), while Texas had the highest at 76.7%.

**Table 2.** Demographic variables of overall sample and by state (Florida, Texas, and Michigan) [1].

| | | Mean (S.D.) or % | | | | |
| --- | --- | --- | --- | --- | --- | --- |
| | | | By State | | | |
| Demographic Variables | | Total Sample | FL | TX | MI | Statistic, *p*-Value |
| (Categorical) | | *N* = 288 | *N* = 80 | *N* = 106 | *N* = 102 | |
| Gender (M = 0; F = 1) | Male | 32.22% | 36.71% | 30.09% | 28.00% | $\chi^2$ = 291.49, 0.000 |
| | Female | 68.61% | 63.29% | 69.91% | 72.00% | |
| Ethnicity (White = 0; Not White = 1) | White | 67.92% | 78.75% | 50.00% | 70.59% | $\chi^2$ = 2500.00, 0.000 |
| | Not White | 32.08% | 21.25% | 50.00% | 29.41% | |
| Ethnicity | White | 67.92% | 78.75% | 50.00% | 70.59% | $F$ = 5800.00, 0.0000 |
| | African American | 2.93% | 1.25% | 6.60% | 1.96% | |
| | Hispanic | 7.33% | 6.25% | 18.87% | 0.98% | |
| | Asian | 14.09% | 6.25% | 15.09% | 19.61% | |
| | Other | 6.03% | 6.25% | 7.55% | 4.90% | |
| Education (Less than = 0; Greater than = 1) | Less than 4 yr. degree plus | 42.52% | 45.57% | 23.30% | 52.00% | $\chi^2$ = 2400.00, 0.0000 |
| | 4 yr. college degree or more | 57.48% | 54.43% | 76.70% | 48.00% | |
| Education | Less than High School | 0.00% | 0.00% | 0% | 0.00% | $F$ = 4800.00, 0.0000 |
| | High school or GED | 7.81% | 10.13% | 2.91% | 9.00% | |
| | Some college | 27.66% | 24.05% | 15.53% | 38.00% | |
| | 2-year college degree | 7.06% | 11.39% | 4.85% | 5.00% | |
| | 4-year college degree | 29.49% | 31.65% | 33.98% | 25.00% | |
| | Master's degree | 20.35% | 21.52% | 29.12% | 14.00% | |
| | Doctoral degree | 5.81% | 1.27% | 9.71% | 7.00% | |
| | Professional Degree (JD, MD) | 1.83% | 0.00% | 3.88% | 2.00% | |
| Age (years old) | | 60.76 (17.73) | 47.71 (15.29) | 60.84 (17.94) | 71.29 (11.19) | $F$ = 25,000.00, <0.0001 |
| Adults in HH (18 or over) | | 2.46 (1.115) | 2.22 (0.94) | 2.58 (1.31) | 2.05 (0.35) | $F$ = 2800.00, 0.000 |
| Children in HH (under 18) | | 0.33 (0.67) | 0.46 (0.82) | 0.33 (0.66) | 0.24 (0.51) | $F$ = 1100.00, 0.000 |
| HH Income (USD $, 000) | | 70.51 (5.44) | 70.38 (5.12) | 63.59 (5.14) | 74.70 (5.83) | $F$ = 3600.00, 0.000 |
| No. Plant Types Purchased | | 2.66 (01.79) | 3.29 (1.91) | 2.66 (1.74) | 2.17 (1.56) | $F$ = 5300.00, 0.000 |
| Plant Knowledge Quiz (0 min, 10 max) | | 4.47 (2.72) | 5.28 (2.79) | 4.33 (2.91) | 3.97 (2.36) | $F$ = 550,000.00, 0.000 |
| Spent on Plants (USD) | | 123.02 (127.71) | 176.73 (132.15) | 113.40 (127.79) | 85.49 (108.14) | $F$ = 9300.00, 0.000 |

[1] Data analyses were generated using the CHISQ and TESTP options in the TABLES statement of the FREQ procedure of SAS Software (SAS for Windows, v 9.4, SAS Institute Inc.). FL = Florida, TX = Texas, MI = Michigan.

The overall mean age of the subjects was 60.7 years, with Florida having the youngest group (47.7) and Michigan the oldest (71.3). The mean household size for the overall sample was 2.5 adults, with Texas having the most adults (2.6), followed by Florida (2.2) and Michigan (2.1). On average, there were 0.33 children per household, with Florida households having the most children (0.50), followed by Texas (0.33) and Michigan (0.2). Average household income in 2018 was $70,510 (S.D. = $5440). Michigan respondents were the most affluent, with an average income of $74,700 (S.D. = $5830), followed by Florida at $70,380 (S.D. = $5120) and Texas at $63,590 (S.D. = $5140).

Researchers asked subjects if they had purchased plants from any of these 12 categories: annuals, perennials, herbs, vegetables, flowering shrubs, evergreen shrubs, fruit-producing trees, evergreen trees, shade trees, indoor flowering potted plants, indoor foliage plants, and succulents in the six months prior to the survey. The subjects, on average, purchased 2.66 different types of plants. Floridians purchased a greater variety of plants, with an average of 3.29 different plant types, compared to Texans (2.66) and Michiganders (2.17). The top three plant types that subjects purchased were annuals, herbs, and vegetables. The average spending was $123.02 for the overall sample. Florida participants spent the most at $176.73 in 2018 on plants and plant-related equipment, while Texans spent $113.40 and Michiganders spent $85.49.

Responses to the actual plant knowledge quiz (Supplementary Materials S7) were normally distributed in the overall sample, with a mean score of 4.47 correct responses out of 10 questions. Floridians performed the best on the quiz with 5.28 correct responses, followed by Texans with 4.33 and Michiganders with 3.97.

*3.2. Conjoint Findings*

Overall, plant type was the most important attribute among the four tested, followed by plant guarantee, water message, and, lastly, price (Table 3). Within plant type, annuals were preferred over woody perennial shrubs. Twenty-five percent off the regular price was the most preferred price, followed by buy-3-get-1-free and the regular price. Of the five plant guarantee levels, a one-year guarantee was the most preferred, followed by six months, 90 days, 30 days, and, lastly, no guarantee. Requiring irrigation was least preferred compared to the no required irrigation message.

We found some differences when comparing responses by state. Michigan and Texas subjects rated plant type as the most important attribute, whereas Floridians rated plant guarantee most important. Price was consistently the least important attribute across all three states. Plant guarantee was consistently rated higher than the water message or price for all three states. Comparatively, the water message was rated higher in relative importance for Florida subjects compared to respondents from Michigan and Texas.

There were some differences by state for the product attribute utility scores. For example, respondents from all three states valued annuals over woody perennial shrubs, but Michiganders derived higher utility from annuals compared to Texans. The mean utility score for the water message also varied by state. Floridians and Texans valued "needs no irrigation" more highly compared to Michiganders. While the price reflected by 25% off and Buy-3-Get-1-Free (B3G1) were identical, Florida subjects had a higher mean utility score for the 25% off, and Michiganders had the lowest mean utility score. Texans had a higher mean utility score for B3G1, while Floridians had the lowest mean utility score. For plant guarantees, Floridians had the highest mean utility score for the one-year guarantee. Michiganders had a relatively higher utility score for the six-month guarantee, while both southern states disliked it, indicated by the negative value. Texans and Floridians liked the 90-day guarantee, while Michiganders disliked the 90-day guarantee, as indicated by the negative utility value. Floridians had the lowest utility score for the blank guarantee message, indicating they did not like it, though respondents from all three states prefer a guarantee of some kind.

To gain further insight into the subjects' decision-making process, the researchers assessed the three eye-tracking metrics and the subject's likelihood-to-buy (LTB) rating for each of the attribute levels (Table 4). Within the plant type attribute, annual plants had a shorter TTFF, meaning they were

seen faster than the woody perennial shrubs. Annual plants had a greater mean TFD and LTB rating, however, there was no difference in FC by the type of plant. The water message attributes had a similar mean TTFF, TFD, and FC, which indicated no visual differences; however, "needs no irrigation" had a greater mean LTB rating. Price attributes differed by TTFF, TFD, FC, and LTB. The B3G1 and 25% off price levels had the same mean TTFF and TFD, and were greater than regular price, meaning that the regular price was noticed faster than the sale prices. The 25% off price level had the greatest mean FC, followed by B3G1 and, lastly, regular price. FC can be an indication of cognition, indicating that participants thought about the 25% off price more than the other price levels. The LTB ratings were similar for both 25% off and B3G1 free and greater than the regular price. For plant guarantee levels, six-month had the greatest TTFF, followed by blank guarantee, one-year, and 90-day. The 30-day had the lowest TTFF. The 30-day guarantee had the greatest TFD, followed by 90-day, one-year, blank, and, lastly, six-month guarantee. FC was no different among the guarantee levels.

**Table 3.** Conjoint analysis showing mean relative importance scores and standard errors (S.E.) for each attribute overall and by state [1].

| Attribute | | Mean (S.E.) Relative Importance | | | | |
|---|---|---|---|---|---|---|
| | | All | By State | | | |
| | | Subjects | FL | TX | MI | (DF) *F*, *p* |
| | | *N* = 288 | *N* = 80 | *N* = 106 | *N* = 102 | |
| Plant Type | | 30.3645 (0.11) | 26.07 (0.15) | 29.14 (0.24) | 35.01 (0.18) | (1) 1254.74, 0.000 |
| Water Message | | 22.8684 (0.09) | 25.49 (0.15) | 24.79 (0.20) | 18.82 (0.13) | (1) 1035.02, 0.000 |
| Price Type | | 19.405 (0.07) | 19.60 (0.11) | 19.50 (0.14) | 19.15 (0.11) | (1) 8.40, 0.0037 |
| Plant Guarantee | | 27.3617 (0.07) | 28.84 (0.11) | 26.57 (0.14) | 27.02 (0.11) | (1) 107.88, 0.000 |
| Attribute | Level | Mean (S.E.) Utility Score | | | | |
| | | All | By State | | | |
| | | Subjects | FL | TX | MI | (DF) *F*, *p* |
| Plant Type | Annual | 0.8678 (0.0050) | 0.8604 (0.0085) | 0.6141 (0.0107) | 1.0319 (0.0079) | (1) 15.96, 0.000 |
| | Shrub | −0.8678 (0.0050) | −0.8604 (0.0085) | −0.6141 (0.0107) | −1.0319 (0.0079) | (1) −15.96, 0.000 |
| Water Message | Needs Irrigation | −0.7002 (0.0040) | −0.8469 (0.0079) | −0.7327 (0.0084) | −0.5648 (0.0050) | (1) 938.49, 0.000 |
| | Needs No Irrigation | 0.7002 (0.0040) | 0.8469 (0.0079) | 0.7327 (0.0084) | 0.5648 (0.0050) | (1) 938.49, 0.0000 |
| Price Type | Regular | −0.438 (0.0030) | −0.4750 (0.0055) | −0.5106 (0.0054) | −0.3638 (0.0045) | (1) 287.84, 0.000 |
| | Buy-3-Get-1-Free | 0.1365 (0.0029) | 0.0833 (0.0057) | 0.2448 (0.0050) | 0.1108 (0.0043) | (1) 7.20, 0.0073 |
| | 25% Off | 0.3015 (0.0024) | 0.3917 (0.0046) | 0.2659 (0.0040) | 0.2530 (0.0037) | (1) 585.36, 0.000 |
| Plant Guarantee | Blank | −0.2334 (0.0026) | −0.3538 (0.0051) | −0.3167 (0.0046) | −0.0846 (0.0035) | (1) 2207.40, 0.000 |
| | 30-Day | −0.4389 (0.0037) | −0.4246 (0.0070) | −0.2914 (0.0070) | −0.5422 (0.0053) | (1) 228.45, 0.000 |
| | 90-Day | 0.0324 (0.0033) | 0.0629 (0.0073) | 0.1345 (0.0053) | −0.0552 (0.0043) | (1) 614.70, 0.000 |
| | 6 Month | 0.0606 (0.0031) | −0.0038 (0.0060) | −0.0289 (0.0051) | 0.1670 (0.0047) | (1) 614.70, 0.000 |
| | 1 Year | 0.5783 (0.0037) | 0.7192 (0.0072) | 0.5025 (0.0072) | 0.5150 (0.0050) | (1) 525.17, 0.000 |

[1] Regression; FL = Florida, TX = Texas, MI = Michigan.

**Table 4.** Eye-tracking attributes by conjoint attribute levels [1].

| Plant | Message | Price | Guarantee | Location | TTFF (sec) Mean (S.D.) | | (DF) *F, p* | TFD (sec) Mean (S.D.) | | (DF) *F, p* | FC Mean (S.D.) | | (DF) *F, p* | LTB Mean (S.D.) | | (DF) *F, p* |
|---|---|---|---|---|---|---|---|---|---|---|---|---|---|---|---|---|
| Annual | | | | | 2.78 (3.64) | a | (1) 12.96, 0.0003 | 1.13 (1.79) | a | (1) 5.03, 0.0249 | 4.85 (7.30) | a | (1) 2.09, 0.1484 | 5.73 (2.54) | a | (1) 2664.27, 0.000 |
| Shrub | | | | | 2.92 (3.86) | b | | 1.09 (1.79) | b | | 4.75 (7.47) | a | | 4.45 (2.62) | b | |
| | Needs No | | | | 2.84 (3.76) | a | (1) 0.00, 0.9709 | 1.11 (1.80) | a | (1) 0.24, 0.6223 | 4.81 (7.42) | a | (1) 0.01, 0.9131 | 5.72 (2.62) | a | (1) 2573.32, 0.000 |
| | Needs | | | | 2.85 (3.74) | a | | 1.11 (1.78) | a | | 4.80 (7.35) | a | | 4.46 (2.54) | b | |
| | | Regular | | | 2.66 (3.62) | b | (2) 2.68, 0.000 | 0.98 (1.61) | b | (2) 81.57, 0.000 | 4.42 (6.91) | c | (1) 30.59, 0.000 | 4.63 (2.65) | b | (2) 355.49, 0.000 |
| | | 25% Off | | | 2.94 (3.85) | a | | 1.19 (1.92) | a | | 5.10 (7.85) | a | | 5.37 (2.62) | a | |
| | | B3G1 | | | 2.95 (3.77) | a | | 1.17 (1.83) | a | | 4.89 (7.34) | b | | 5.33 (2.64) | a | |
| | | | Blank | | 2.89 (3.84) | b | (4) 3.39, 0.0089 | 1.12 (1.84) | b | (4) 2.84, 0.0229 | 4.83 (7.54) | a | (4) 3.53, 0.0068 | 4.93 (2.64) | c | (4) 392.03, 0.000 |
| | | | 30-Day | | 2.74 (3.64) | a | | 1.15 (1.81) | b | | 4.91 (7.17) | a | | 5.54 (2.51) | b | |
| | | | 90-Day | | 2.77 (2.57) | a | | 1.13 (1.79) | b | | 4.86 (7.42) | a | | 5.93 (2.56) | a | |
| | | | 6 Month | | 2.95 (3.68) | b | | 1.05 (1.77) | a | | 4.49 (7.36) | b | | 4.40 (2.55) | e | |
| | | | 1 Year | | 2.85 (3.92) | ab | | 1.11 (1.70) | ab | | 4.85 (7.26) | a | | 4.84 (2.74) | d | |
| | | | | Left | 4.42 (4.31) | a | (2) 328.09, 0.000 | 0.66 (.80) | c | (2) 176.04, 0.000 | 2.81 (3.36) | c | (1) 246.99, 0.000 | 5.13 (2.67) | b | (2) 20.51, 0.000 |
| | | | | Middle | 2.01 (3.30) | c | | 1.10 (1.08) | a | | 4.92 (4.45) | a | | 5.35 (2.67) | a | |
| | | | | Right | 3.26 (3.45) | b | | 0.91 (1.00) | b | | 3.82 (3.87) | b | | 4.93 (2.64) | c | |

[1] Different letters across columns indicate significant differences of means at $p < 0.05$; ANOVA analysis; TTFF = time to first fixation; TFD = total fixation duration; FC = fixation count; LTB = likelihood to buy.

The analysis of whether the water message information was viewed differently by sign location in the display showed that the left sign had the highest TTFF and the middle sign location had the lowest TTFF. The middle sign location had the longest TFD and the most FC, followed by the right sign location and, lastly, the left. The middle sign location was rated the highest in LTB, followed by the right sign location and, lastly, the left sign location. In essence, the water message information viewing was the same by the reaction time, but different in the rating of LTB. Yet, the location of the water information led to differences in viewing and LTB, with the highest LTB elicited for the middle location.

We wanted to understand the proportion of consumers who ignored the water message. We compared the percentage of people who had TTFF on the water message sign equal to zero, meaning they did not look at the sign. For Florida, the range of subjects who ignored the water sign ranged from 3.75% to 32.5%. For Texas, the range of subjects who ignored the water sign ranged from 0.9% to 2%. For Michigan, the range of subjects who did not look at the water message sign ranged from 0% to 14.71%. With the exception of one subject in Florida, less than 15% of all subjects ignored the water message

### 3.3. Principal Component Analyses

Results for the principal component analysis of 29 landscape and plant importance items, adapted from Syme et al. [63] and Behe et al. [9], yielded two constructs. We labeled them as Active Landscape Enjoyment and Landscape Aesthetic (Supplementary Materials S8), which were similar to the previously published scales. The second principal component analysis was used on plant expertise and involvement items, as used in Behe et al. [9]. One component that emerged was called Plant Expertise (Supplementary Materials S8). When comparing the mean scores for these components across states, Texans found the greatest active landscape enjoyment, followed by Floridians and, lastly, Michiganders (Table 5). Floridians felt the strongest about their landscape aesthetic and plant expertise, followed by Texans and, lastly, Michiganders.

**Table 5.** Mean comparisons of the principal components by state [1].

| Component | Florida | | Texas | | Michigan | | F, p |
|---|---|---|---|---|---|---|---|
| | | | **Means Comparison** Means (S.D.) | | | | |
| Active Landscape Enjoyment | 0.0126 (1.0789) | b | 0.0543 (1.0063) | c | −0.1378 (0.9245) | a | 151.20, 0.000 |
| Landscape Aesthetic | 0.1487 (1.0117) | c | −0.0486 (1.0304) | b | −0.1292 (0.9571) | a | 309.14, 0.0000 |
| Plant Expertise | 0.3399 (1.0247) | c | −0.0182 (1.0147) | b | −0.2825 (0.8731) | a | 1560.45, 0.000 |

[1] Different letters across columns indicate significant differences of means at $p < 0.05$.

### 3.4. Conjoint Clusters

Three clusters emerged from the cluster analysis, and were compared using ANOVA analysis of the demographic characteristics of the clusters, including gender, age, ethnicity, household number of adults and children, education level, income, and expenditures on plant-related products in 2016 (Table 6).

**Table 6.** Cluster proportions including PCA and part-worth utility values [1].

| Variable | Big Spenders N = 167 | | Ambivalents N = 43 | | Plant Buyers N = 78 | | *p*-Value |
|---|---|---|---|---|---|---|---|
| | | | Mean (S.D.) | | | | |
| Age | 60.49 (18.08) | a | 61.53 (16.57) | b | 61.31 (17.46) | b | 0.0000 |
| Gender (% Female) | 0.69 (0.46) | b | 0.72 (0.45) | c | 0.65 (0.48) | a | 0.0000 |
| White (% non-white) | 0.32 (0.46) | a | 0.36 (0.48) | b | 0.31 (0.46) | a | 0.0000 |
| 4 yr. college degree or more (% with) | 0.53 (0.50) | a | 0.65 (0.48) | c | 0.63 (0.48) | b | 0.0000 |
| Adults in HH (≥18 years) | 2.54 (1.25) | c | 2.24 (1.09) | a | 2.45 (0.93) | b | 0.0000 |
| Children in HH (>18 years) | 0.33 (0.67) | b | 0.24 (0.56) | a | 0.39 (0.74) | c | 0.0000 |
| HH Income (USD $, 000) | 73.93 (54.98) | a | 70.48 (62.50) | c | 62.80 (47.05) | b | 0.0000 |
| No. Plant Types Purchased | 2.86 (1.90) | c | 2.16 (1.74) | a | 2.53 (1.48) | b | 0.0000 |
| Spent on Plants (USD) | 135.19 (128.15) | c | 82.37 (114.05) | a | 118.18 (129.40) | b | 0.0000 |
| Plant Knowledge Quiz | 4.46 (2.68) | a | 4.42 (2.94) | a | 4.60 (2.64) | b | 0.0000 |
| Active Landscape Enjoyment | 0.04 (0.97) | b | −0.13 (1.13) | a | −0.14 (0.98) | a | 0.0000 |
| Landscape Aesthetic | 0.027 (0.98) | c | −0.099 (1.06) | a | −0.062 (1.00) | b | 0.0000 |
| Plant Expertise | 0.048 (1.02) | c | −0.264 (1.13) | a | −0.008 (0.83) | b | 0.0000 |
| Plant Relative Importance | | | | | | | |
| Annual | 0.59 (0.86) | b | 2.66 (0.78) | c | 0.33 (0.60) | a | 0.0000 |
| Shrub | −0.59 (0.86) | b | −2.66 (0.78) | c | −0.33 (0.60) | a | 0.0000 |
| Message Relative Importance | | | | | | | |
| Needs | −0.29 (0.41) | c | −0.39 (0.48) | b | −1.76 (0.81) | a | 0.0000 |
| Needs No | 0.29 (0.41) | a | 0.39 (0.48) | b | 1.76 (0.81) | c | 0.0000 |
| Price Relative Importance | | | | | | | |
| Regular Price | −0.52 (0.70) | a | −0.27 (0.46) | c | −0.40 (0.45) | b | 0.0000 |
| Buy-3-Get-1-Free | 0.19 (0.61) | c | 0.02 (0.62) | a | 0.14 (0.56) | b | 0.0000 |
| 25% Off | 0.32 (0.53) | b | 0.26 (0.45) | a | 0.26 (0.45) | a | 0.0000 |
| Guarantee Relative Importance | | | | | | | |
| Blank | −0.27 (0.54) | a | −0.10 (0.51) | c | −0.26 (0.52) | c | 0.0000 |
| 30-Day | −0.33 (0.69) | c | −0.80 (0.94) | a | −0.39 (0.77) | b | 0.0000 |
| 90-Day | 0.12 (0.54) | b | −0.50 (0.93) | c | 0.20 (0.63) | a | 0.0000 |
| 6 Month | −0.02 (0.56) | b | 0.48 (0.70) | c | −0.05 (0.66) | a | 0.0000 |
| 1 Year | 0.51 (0.70) | a | 0.92 (0.94) | b | 0.50 (0.77) | a | 0.0000 |

[1] Different letters across columns indicate significant differences of means at *p* < 0.05. HH = household; y indicates $X^2$; z indicates GLM means.

Cluster 1 members were the youngest among the clusters, intermediate in terms of the percentage of women and non-white respondents, had the least education, the most adults in the household, and had the second most children in the household. Members of Cluster 1 also purchased the largest variety of plants (mean of 2.86 plant categories), spent the most on plant-related purchases, and had the highest plant expertise, landscape pride, and aesthetic mean scores compared to the other two clusters. Researchers labeled Cluster 1 as "Big Spenders". Big Spenders least preferred the water message of "needs no irrigation", preferred the 25% off price more than the other two clusters, and preferred one-year guarantees, followed by 90-day, six-month, 30-day, and no plant guarantees.

Members of Cluster 2 were older and had a higher percentage of women. This cluster was intermediate in the percentage of non-white respondents. Members of Cluster 2 were the most educated, had highest income, and had the least number of children and adults in the household. Overall, members of Cluster 2 had the lowest score on the plant knowledge quiz, spent the least on plants and plant-related expenditures, and purchased the lowest diversity of plants (2.16 categories). They had slightly negative scores on landscape pride and aesthetics, as well as plant expertise. Because Cluster 2 was the least engaged of the three clusters, they were labeled "Ambivalents". Of all of the clusters, Ambivalents most preferred annual plants, preferred 25% off above B3G1 and regular price, and preferred one-year and six-month plant guarantees.

Cluster 3 consisted of a higher percentage of males and children per household. Their plant purchase diversity was similar to Big Spenders, but was higher than Ambivalents. Members of this cluster were intermediate in the percentage of White respondents, but similar in age to Ambivalents.

Because they scored the highest on the actual plant knowledge quiz (4.55), they were labeled "Plant Buyers." However, the mean score for landscape pride, aesthetics, and plant expertise were the lowest of the clusters, indicating that this group was not as interested in landscape aesthetics or pride compared to the other two consumer groups. Additionally, Plant Buyers rated the water message attribute of "needs no irrigation" higher than the other two clusters, but cared about the plant type the least. Plant Buyers also liked the 90-day and one-year guarantees more than six-month, 30-day, and no guarantee.

To evaluate the clusters further, their part-worth utility values derived from the conjoint analysis were compared by cluster using the three eye-tracking metrics: TTFF, TFD, and FC and LTB ratings (Table 7). For the water message attribute levels, there were no differences in the TTFF for the three clusters between the message "Needs no irrigation" and "Needs irrigation". Additionally, the TFD was similar for both messages and across all clusters. The LTB ratings for the water messages were higher for the "needs no irrigation" for all three clusters when compared to the "needs irrigation" water message. Plant Buyers ranked the "needs no irrigation" message the highest LTB, followed by Big Spenders and then Ambivalents. Big Spenders ranked "needs irrigation" the highest of the clusters, followed by Ambivalents and, lastly, Plant Buyers.

**Table 7.** Conjoint clusters eye-tracking for water messaging attributes [1].

| Water Message Mean (S.D.) | Big Spenders N = 167 | | | | Ambivalents N = 43 | | | | Plant Buyers N = 78 | | | |
|---|---|---|---|---|---|---|---|---|---|---|---|---|
| | TTFF | TFD | FC | LTB | TTFF | TFD | FC | LTB | TTFF | TFD | FC | LTB |
| Needs No | 2.85 (1.80) ab | 1.12 (1.80) a | 4.81 (7.47) b | 5.63 (2.57) b | 2.94 (1.89) b | 1.07 (1.89) a | 4.61 (7.32) a | 5.37 (2.91) d | 2.80 (1.78) ab | 1.13 (1.78) a | 4.95 (7.38) b | 6.19 (2.50) e |
| Needs | 2.89 (1.76) b | 1.12 (1.76) a | 4.81 (7.38) b | 4.71 (2.48) c | 2.96 (1.86) b | 1.08 (1.86) a | 4.63 (7.22) a | 4.18 (2.73) a | 2.72 (1.78) a | 1.13 (1.78) a | 4.91 (7.41) b | 4.12 (2.54) a |

[1] Lower case letters indicate differences among metrics in each cluster (TTFF, TFD, FC, and LTB). TTFF = time to first fixation; TFD = total fixation duration; FC = fixation count; LTB = likelihood to buy.

## 4. Discussion

### 4.1. Demographic Characteristics

Our sample had demographic similarities with a large sample of US lawn and garden consumers [64]. However, without their variances (only means were published), researchers were unable to make statistical comparisons. The sample in this study had >60% female participants, which was greater than the percentage of women in the US, as reported by the US Census (51%) [65], but was consistent with plant purchasers as described by the National Gardening Association [64]. Additionally, we had a greater representation of people of color in comparison to the US Census, where the total population is 75% white, 14% African American, 6.6% Asian, and 11% other ethnicities. Eighteen percent of the US population is Hispanic/Latino. The sample also included more individuals who had a higher level of education. This was greater than the 36% of United States citizens who hold a bachelor's degree or higher. Household income here was comparable, where the median household income in 2018 was $63,179. The size of the family unit was comparable to the US Census; there were 2.6 persons per household and 3.24 persons per family unit. Thus, the sample for the present study was consistent with a national sample of lawn and garden product consumers, with the only difference being that the average spending in lawn and garden was $123.02 for the overall sample, which was below the national average spending of $503 in lawn and gardening activities per household by the National Gardening Survey [64].

Demographically, Texans were the most ethnically diverse and were more educated, but had the lowest income. This ethnic diversity could be reflective of the average Texas plant consumer due to the high Hispanic population present in the state. Floridians were the youngest sample, yet spent the most on plants and plant-related supplies—double what Michiganders spent—and spent one-third more than Texans. Additionally, they were the most knowledgeable about plants, shown by their score on

the knowledge quiz. Perhaps this result was due to the warmer climate in Florida, allowing a longer growing season and greater diversity of plants that can be cultivated.

### 4.2. Conjoint Findings

The conjoint study produced findings consistent with prior research [8,44,66,67]. Overall, plant type was the most important product attribute, while price was the least important. Floridians had a higher mean importance rank for plant guarantees and water messaging, indicating that those attributes weighed heavily in their purchase intention. This may have arisen from past drought events causing plant death and a desire for some potential recourse in the form of a product guarantee. Michiganders had a low relative importance for the water message, which may be due to a history of relatively minor drought conditions (level D0–D4) in the past 20 years [68].

### 4.3. Part-Worth Conjoint Utility Values and Visual Attention

Delving into the part-worth utility values, annual plants were more preferred than shrubs. Perhaps this is due to the attractive flower color on the annuals, which shrubs used in this study did not have. Annual plants are also purchased by a greater number of US consumers compared to shrubs [64], so more Americans could be more familiar with annuals than with shrubs. Annual plants captured visual attention faster (lower TTFF) compared to the shrubs, and subjects looked at the annuals longer (greater TFD), indicating visually that they preferred the colorful, flowering annuals to shrubs. There was no difference in the overall visual attention to the water messaging information, yet there was a difference in the LTB rating by message type. Because the messages varied by only one word, we postulate that they required a similar time frame to cognitively process the information. Therefore, both of the water messages resulted in the same gaze behavior. Both Texas and Florida have historically experienced moderate to severe drought, so respondents from these states responded to the water messages differently than Michigan respondents. Florida subjects were most responsive to the "needs no irrigation" water message. Texas also had a high mean importance rating for the "needs no irrigation" water message, despite not having a drought condition at the time of data collection. Participants from those states may seek plants that are more drought tolerant to improve the odds of establishment and survival.

Sign location had an impact on visual attention. Specifically, the left-hand sign was looked at the fastest, which is logical, considering English readers move their eyes from left to right. Consistent with the Central Gaze Theorem [69], subjects viewed the central sign location the longest, returned to the location, and rated the water message the highest LTB when it was in the middle location, despite using bull's eyes to prevent the bias effect of fixating on the center of the screen. Therefore, positioning relevant information the retailer wishes to communicate to the buyer in a middle location may result in a longer viewing time, potentially greater cognition, and, ultimately, greater chance of purchase.

A regular or non-sale price has a trade-off value with a sale price. While a regular price would gain the firm the most capital, not surprisingly, consumers were more likely to buy with discounts present. Overall, 25% off and B3G1 were more preferred price levels than regular price. Both are the same discount, but in past literature, 25% off was more preferred than B3G1 [29]. For the B3G1 discount, subjects may experience more cognitive load by mentally calculating the discount, therefore leading to longer time to fixation. The regular price attribute had the shortest TTFF, which is consistent with past literature, where people found the highest price more quickly [36]. Additionally, subjects fixated on their preferred choices more than their non-preferred choice [29].

Plant guarantees are commonly used by big box (mass) retailers, but are less commonly offered by independent garden centers [43]. The findings presented here indicate that the absence of a guarantee detracted from the value of the plant, while a one-year guarantee contributed positively to purchase intention. This value added could be attributed to a perception of risk mitigation with a guarantee present. Yet, there were geographic differences for the length of guarantee preference between none and one-year. The attribute importance of plant guarantees was ranked as longer

guarantees being preferred over shorter guarantees. Visual attention to 30-day and 90-day guarantees was more pronounced, with these guarantees having the shortest TTFF, the longest TFD, and the highest FC. These findings might be associated with the experimental design, as the annual plants were not paired with a guarantee over 90 days because, intuitively, annuals will not live longer than a season. Additionally, the increased visual attention to the 30- and 90-day guarantees could reflect an internal decision-making process to determine if this guarantee is long enough to justify the plant purchase. Is the risk worth the reward? Respondents from both southern states, Florida and Texas, preferred either one-year or 90-day guarantees; six-month guarantees detracted from the value (had a negative utility score). Those participants may have been looking for plants that will survive the season and/or have long-term establishment. For example, they may not want to purchase a plant in February and return the plant in July when the season is unfavorable to new plantings. Michiganders, on the other hand, did not want shorter guarantees, including 90-days, but preferred six-month and one-year guarantees. Preferring longer guarantees could be reflective of the plant's survival over the winter season. The utility values per length of guarantee did not demonstrate a linear relationship for the three state sub-samples. Thus, future research should include a wider geographic representation, perhaps by drought likelihood or length of growing season; a focus on assessing the relationship of plant types, customer expectations, and guarantee length should also be investigated.

*4.4. Conjoint Clusters*

"Big Spenders" tended to be more representative of Generation X (individuals born between 1965 and 1980) and older Generation Y individuals (individuals born between 1981 and 1995). These individuals report enjoying being around plants and have a lot of plant expertise compared to the other two clusters. "Ambivalents", like the "No Landscape Enjoyment" Cluster in Knuth et al. [9], scored low on the knowledge quiz, did not have as much plant expertise or enjoy landscape activities, and purchased the fewest plant types. Ambivalents also were the most ethnically diverse. This result is similar to the findings of Dennis et al. [70], and indicates that the green industry could increase communication with more ethnically diverse consumers.

The Plant Buyers were closer to a typical consumer of green industry products [64]. Though the typical plant buyer has a higher income [64], the "Plant Buyer" did not, and had an income closer to the US median income level. Interestingly, these individuals rated their plant expertise lower compared to the other clusters, but had the highest plant knowledge score. This finding might represent the hierarchy of competence. The hierarchy of competence construct reflects an awareness of knowledge deficiencies and, therefore, may be a more accurate indication of real plant knowledge than self-reported knowledge (plant expertise construct). Plant Buyers greatly valued the water message attribute. This coincided with past literature, where water conservers usually were more likely female, in lower income households, and were plant buyers [11,12]. Plant Buyers' and Big Spenders' visual attention to the "needs no irrigation" message was similar, yet the Plant Buyers rated their LTB greater when this sign was present. The similarity in visual attention to the water messages may indicate a similar level of cognitive effort to make a decision, and there may be pre-established attitudes regarding the importance of water conservation. This finding corresponds with past literature that shows that homeowners actively interested in water conservation are also interested in plant-related activities [10].

## 5. Conclusions

Study limitations include a limited number of data collection sites and sample sizes, largely due to funding limitations. Only two types of plants were investigated to reduce participant fatigue. The same limitations apply to water messages and guarantee levels. The present study identified the effects that these product attributes have on purchase intentions. More investigation is merited on more water conserving messages and guarantee levels.

Because homeowners are more concerned with long-term drought effects than other weather patterns, including heavy rains [6], we sought to understand how a water conservation message in the

retail environment affects consumers' purchase intentions for outdoor plants. Highly visible landscape, lawn, and garden irrigation is frequently a target of mandatory water restrictions and regulation. Our research question is important, because sales of ornamental plants could be reduced if water restrictions prevent consumers from irrigating plants outdoors. Plant selection becomes critical for purchase first and survival in the landscape second. However, simply identifying species and cultivars that require less water is insufficient to stimulate sales if consumers are unaware of that attribute or ignore the information. Developing effective communications (e.g., signage) and identifying groups of consumers who respond positively to water conserving messaging may help growers and retailers to continue to sell landscape plants during drought conditions. Findings from the current study contribute to hopeful future sales in several ways.

We examined a combination of a water conservation message, plant guarantee, and price on purchase intention of live plants in a retail display. Firstly, and consistent with prior research, we found that plant type was the most important attribute in the purchase intention [8,9,31,32,37,66,67]. Second, and more importantly, for some consumers in states that historically experienced drought, a water saving message also played an important role in purchase intention. This supports the notion that some consumers are concerned with drought conditions and react to them [6]. Retail garden centers in drought-prone areas may consider including water conserving messages in point-of-purchase materials to bring attention to plants requiring less water for establishment and survival. Even retailers in non-drought areas may benefit slightly from water-conserving messages, because some consumers may carry an awareness of drought effecting other landscape plants.

Third, providing water requirement information in a visible manner makes the information easier to find and has the potential to increase the likelihood that the consumer will look at the information and take it into account during their decision. Prior work showed that plant purchasers who paid more attention to water saving information on signage had a greater likelihood to buy a plant [22]. Our work supports this supposition, since approximately 85% of the subjects in the current study viewed the water saving message. We also demonstrated that this plant feature was able to attract consumers' attention, as well as positively influenced purchase intention, consistent with prior research [22].

Fourth, results for the relative importance of plant guarantees contributed to the literature in that the mere presence of those guarantees enhanced the likelihood to purchase that plant. Plant guarantees provide potential recourse for the consumer if products fail, such as a plant dying from lack of irrigation [42–44]. Here, plant guarantees were more important than both the water message and price, indicating that it is a salient feature in many consumers' purchase intentions. Although fewer independent plant retailers than big box retailers offer plant guarantees, providing a longer-term guarantee on plant purchases mitigates some of the perceived risk for consumers, which may be amplified in drought situations. However, it should be noted that guarantees for annuals would not logically exceed 90 days.

Consumers are rarely homogenous in their attitudes, preferences, and perceptions. More young consumers, younger Generation X and older Generation Y, are purchasing plants and are expressing positive attitudes regarding their landscape behaviors [64]. This should be of great interest to green industry stakeholders, and will require additional investigations to observe if their attitudes and behaviors differ from the "traditional" plant buyer, as defined in Cohen [64] (more likely to be older, female, and affluent). Convincing consumers that the costs of water conservation yields environmental benefits is of utmost importance. Established plant consumers are interested in water conservation and most positively assess water conservation messaging. These consumers would benefit not only from providing information on signage about the plant, but also the necessary watering requirements.

**Supplementary Materials:** The following are available online at http://www.mdpi.com/2073-4441/12/12/3487/s1 and https://osf.io/yvd7g/?view_only=9ddfa79feff14fa2bae7e5dd06633589 (data, conjoint, principal component, and general analysis code).

**Author Contributions:** Conceptualization, B.K.B., C.R.H., P.T.H. and R.T.F.; methodology, B.K.B., C.R.H., P.T.H. and R.T.F.; software, M.J.K., B.K.B.; formal analysis, M.J.K.; investigation, M.J.K. and B.K.B.; resources, B.K.B., C.R.H., and H.K.; data curation, M.J.K.; writing—original draft preparation, M.J.K., B.K.B.; writing—review and editing, P.T.H., C.R.H., R.T.F., H.K.; supervision, B.K.B.; funding acquisition, B.K.B., C.R.H., P.T.H. and R.T.F. All authors have read and agreed to the published version of the manuscript.

**Funding:** This research was funded by USDA SCRI Clean WateR3—Reduce, Remediate, Recycle Grant Number 2014-51181-22372; USDA NIFA Hatch Projects MICL 02085, MICL 1011569, and TEX0-1-7051; Michigan State University AgBioResearch, and MSU Project GREEN and Texas A&M AgriLife Research.

**Acknowledgments:** Thank you to Lynne Sage for providing invaluable assistance in creating the experimental design and collection of data. Thank you to Marco Palma for allowing us to collect data at the Human Behavior Lab at Texas A&M University.

**Conflicts of Interest:** The authors declare no conflict of interest. The funders had no role in the design of the study; in the collection, analyses, or interpretation of data; in the writing of the manuscript, or in the decision to publish the results.

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
