# Peer review of "Water Conserving Message Influences Purchasing Decision of Consumers"

_water, doi:10.3390/w12123487_

Round 1

Reviewer 1 Report

Ethics statement

Please include ethics application reference number for this study on Section 2 under M&M.

Section 2 needs some subsectioning - please separate the materials (e.g. participants) to methods (e.g. stats methods).

Line 211 - A mention of statistical power of at least 75 people were mentioned. Did the authors run any power analysis prior?

Line 226 - the authors mentioned fatigue where participants were shown 36/60 combinations, was there a pilot for this? This still seems a lot.

Line 236 - how long exactly did the participants see these images for? More detail on the experimental protocol is needed. Perhaps the authors can create a flow of experiment in explaining this better?

Line 255 - why was HCA analysis used not other clustering techniques? Is there a reason here?

Line 261 - what was this dimension landscape, plant expertise retrieved from?

Line 267 - Also add KMO to suggest sampling adequacy.

Line 348 - How did the authors actually measure LTB?

Table 4. It seems that posthoc comparison is carried out? If yes, then please add details on methods.

Section 3.4 - why did the authors use non-polynomial approach?

Table 7. The abcs that's suggested by the authors is very difficult to follow. Is there a reason for upper/lower case. If it's just different then please use small case letter.

Section 4. Please subsection the discussion section based on the key results and main highlight.

Reviewer 2 Report

This is a well-written paper on the important topic of how customers perceive signage in garden centers. It also provides relevant information on the importance of price/sale, guarantees, and water conservation importance. 

I have some minor edits/questions for clarifiication:

Line 218: the subjects paid a $10 incentive or were paid a $10 incentive?

225: need irrigation vs. no irrigation? – I don’t understand why no irrigation was used as any new plant is going to need some water for establishment

229: and were appropriate for all three states

290: … highest ethnic diversity with 50% identifying as African American (6.6%) – I believe this is mean that 50% of the sample was ethnically diverse but it reads as 50% identified as African American (which is contradicted by the 6.6%) – please clarify

Table 2: Can this be reformatted to make the data presentation more consistent (some lines all on one line, other lines mixed)

Table 3: again the spacing could be adjusted to make it more clear which line (attribute) the data is corresponding to (ie – with the Price type information)

Line 468-474: less perceived risk with annuals?

Round 2

Reviewer 1 Report

I'd like to thank the authors for addressing the comments.

Just one minor comment is to make sure all the letters for posthocs in the Tables to be consistent, please make all of them small as one Table having caps and the other not. 
